# Southern Ocean latitudinal gradients of Cloud Condensation Nuclei

Ruhi S. Humphries[1,2], Melita D. Keywood[1,2], Sean Gribben[1], Ian M. McRobert[3], Jason P. Ward[1], Paul Selleck[1], Sally Taylor[1], James Harnwell[1], Connor Flynn[4], Gourihar R. Kulkarni[5], Gerald G. Mace[6], Alain Protat[7,2], Simon P. Alexander[8,2], and Greg McFarquhar[4,9]

[1]Climate Science Centre, CSIRO Oceans and Atmosphere, Melbourne, Australia
[2]Australian Antarctic Program Partnership, Institute for Marine and Antarctic Studies, University of Tasmania, Hobart, Tasmania
[3]Engineering and Technology Program, CSIRO National Collections and Marine Infrastructure, Hobart, Australia
[4]School of Meteorology, University of Oklahoma, Norman, United States of America
[5]Atmospheric Sciences and Global Change Division, Pacific Northwest National Laboratory, Richland, United States of America
[6]Department of Atmospheric Science, University of Utah, Salt Lake City, United States of America
[7]Australian Bureau of Meteorology, Melbourne, Australia
[8]Australian Antarctic Division, Channel Highway, Kingston, Tasmania 7050
[9]Cooperative Institute for Mesoscale Meteorological Studies, University of Oklahoma, Norman, United States of America

**Correspondence:** Ruhi S. Humphries (Ruhi.Humphries@csiro.au)

**Abstract.** The Southern Ocean region is one of the most pristine in the world, and serves as an important proxy for the pre-industrial atmosphere. Improving our understanding of the natural processes in this region is likely to result in the largest reductions in the uncertainty of climate and earth system models. While remoteness from anthropogenic and continental sources is responsible for its clean atmosphere, this also results in the dearth of atmospheric observations in the region. Here we present a statistical summary of the latitudinal gradient of aerosol and cloud condensation nuclei concentrations obtained from five voyages spanning the Southern Ocean between Australia and Antarctica from late-spring to early autumn (October to March) of the 2017/18 austral seasons. Three main regions of influence were identified: the northern sector (40-45$^o$S) where continental and anthropogenic sources coexist with background marine aerosol populations; the mid-latitude sector (45-65$^o$S), where the aerosol populations reflected a mixture of biogenic and sea-salt aerosol; and the southern sector (65-70$^o$S), south of the atmospheric Polar Front, where sea-salt aerosol concentrations were greatly reduced and aerosol populations were primarily biologically-derived sulfur species with a significant history in the Antarctic free-troposphere. The northern sector showed the highest number concentrations with median ($25^{th}$ to $75^{th}$ percentiles) $CN_{10}$ and $CCN_{0.5}$ concentrations of 681 (388 - 839) cm$^{-3}$ and 322 (105 - 443) cm$^{-3}$, respectively. Concentrations in the mid-latitudes were typically around 350 cm$^{-3}$ and 160 cm$^{-3}$ for $CN_{10}$ and $CCN_{0.5}$, respectively. In the southern sector, concentrations rose markedly, reaching 447 (298 - 446) cm$^{-3}$ and 232 (186 - 271) cm$^{-3}$ for $CN_{10}$ and $CCN_{0.5}$, respectively. The aerosol composition in this sector was marked by a distinct drop in sea-salt and increase in both sulfate fraction and absolute concentrations, resulting in a substantially higher $CCN_{0.5}$/$CN_{10}$ activation ratio of 0.8 compared to around 0.4 for mid-latitudes. Long-term measurements at land-based research stations surrounding the Southern Ocean were found to be good representations at their respective

latitudes, however this study highlighted the need for more long-term measurements in the region. CCN observations at Cape Grim ($40^o39$'S) corresponded with CCN measurements from northern and mid-latitude sectors, while $CN_{10}$ observations only corresponded with observations from the northern sector. Measurements from a simultaneous two-year campaign at Macquarie Island ($54^o30$'S) were found to represent all aerosol species well. The southern-most latitudes differed significantly from either of these stations and previous work suggests that Antarctic stations on the East Antarctic coastline do not represent the East Antarctic sea-ice latitudes well. Further measurements are needed to capture the long-term, seasonal and longitudinal variability in aerosol processes across the Southern Ocean.

*Copyright statement.* TEXT

## 1 Introduction

Being remote from major population centers and continental influence, the atmosphere of the Southern Ocean represents one of the most pristine on the planet. Because of this, it as an ideal region to understand the pre-industrial atmosphere and the natural processes that are often masked by the much larger signals associated with anthropogenic activity (Carslaw et al., 2013; McCoy et al., 2020). In particular, the Southern Ocean presents a unique test-bed for deepening our understanding of aerosol-cloud interactions and the role of marine biogenic aerosol and their precursors. This is particularly pertinent since the Southern Ocean region exhibits significant uncertainties and biases in the simulations of clouds, aerosols and air-sea exchanges in climate and earth system models (Marchand et al., 2014; Shindell et al., 2013; Pierce and Adams, 2009). These biases can be traced to a poor understanding of the underlying physical processes occurring in the region and can have effects on the global energy budget (Trenberth and Fasullo, 2010), tropical rainfall distributions (Frey and Kay, 2018), and our ability to simulate the impact to global cloud and carbon-cycle feedbacks on climate change (IPCC, 2014; Gettelman et al., 2016).

The remoteness, extreme weather and ocean conditions make *in-situ* observations in this region rare, and until recently, only a handful of aerosol measurements have been made during either transits to Antarctica, or by the few intensive field campaigns focused on the region (Bigg, 1990; Bates et al., 1998; O'Dowd et al., 1997; Boers, 1995; Alexander and Protat, 2019). Having recognised the importance of the Southern Ocean region to the climate and earth system as a whole, the number of campaigns has increased significantly within the last decade and includes HIPPO (HIAPER Pole-to-Pole Observations, 2009 and 2011, aircraft; Wofsy (2011)), SOAP (Surface Ocean Aerosol Production, 2012, vessel; Law et al. (2017)), SIPEXII (Sea Ice Physics EXperiment, 2012, vessel; Humphries et al. (2015, 2016)), PEGASO (Plankton-derived Emissions of trace Gases and Aerosols in the Southern Ocean, 2015, vessel; Dall'Osto et al. (2017); Fossum et al. (2018)), ORCAS (O2/N2 Ratio and CO2 Airborne Southern Ocean, 2016, aircraft; Stephens et al. (2018)), ACE-SPACE (Antartic Circumnavigation Expedition - Study of Preindustrial-like Aerosol Climate Effects, 2017, vessel; Schmale et al. (2019)), ATom (Atmospheric Tomography, 2017, aircraft; Brock et al. (2019)), CAPRICORN (Clouds, Aerosols, Precipitation, Radiation, and atmospherIc Composition Over the southeRn oceaN, 2016 and 2018, vessel), MARCUS (Measurements of Aerosols, Radiation and CloUds over the

Southern Oceans , 2017/18, vessel; Sato et al. (2018)), MICRE (Macquarie Island Cloud and Radiation Experiment, 2016-2018, station) and SOCRATES (SO Clouds, Radiation, Aerosol Transport Experimental Study, 2018, aircraft; McFarquhar et al. (2021); Mace and Protat (2018); Mace et al. (2021). Of particular note are the long-term measurement stations of the Global Atmosphere Watch (GAW) programme, Cape Grim, Tasmania (Ayers et al., 1997; Gras, 1990; Gras and Keywood, 2017), and

the recently (2015) commissioned mobile station, the RV Investigator (Humphries et al., 2021b). The RV Investigator, while continuously undertaking a suite of comprehensive trace-gas, aerosol and cloud measurements in the region, has also hosted a number of intensive field campaigns in the Southern Ocean, such as CWT (2015, IN2015_E01; Alroe et al. (2020)), the maiden voyage (2015, IN2015_V01; Protat et al. (2017)), CAPRICORN (2016, IN2016_V02 and 2018, IN2018_V01), I2E (2016, IN2016_V03; unpublished), and PCAN (2017, IN2017_V01; Simmons et al. (2020)).

Analyses of these recently acquired datasets is ongoing. There is now the opportunity, never before apparent, to combine the multitude of recent measurements from this region into a unified dataset that can be probed to gain deeper insights. In this manuscript, datasets measured during the simultaneous MARCUS and CAPRICORN2 campaigns, are combined and assessed to understand the summertime latitudinal variability across the Australasian sector of the Southern Ocean.

In the mid- and high-latitude Southern Ocean and Antarctic region, aerosols are typically derived from natural sources, in-
cluding primary particles (sea spray and bubble bursting), which makes up the vast majority of the aerosol mass, and secondary particles, which drive the number concentrations of both condensation nuclei (CN) and cloud condensation nuclei (CCN). Early observations of aerosol composition and CN at several Antarctic locations reviewed in Shaw (1988), identified that Antarctic aerosol was dominated by sulfate, and that biological processes, primarily emissions from phytoplankton, were the most likely source of this sulfate. Since then, many studies have shown that secondary particles in the region originate primarily from the
oxidation of dimethyl sulfide (DMS), emitted from phytoplankton, into tertiary volatile compounds such as methanesulfonic acid (MSA) and sulfuric acid which condense to nucleate and grow aerosols (Bates et al., 1998; Covert et al., 1998; Quinn et al., 2000; Rinaldi et al., 2010, 2020; Frossard et al., 2014; Sanchez et al., 2018). These phytoplankton populations, and their subsequent DMS emissions, have significant seasonal cycles (Lana et al., 2011), resulting in major changes in aerosol concentrations throughout the year.

The long term CCN record from Cape Grim demonstrates the clear seasonal cycle in CCN concentrations with maxima occurring during austral summer and minima during the austral winter (Gras, 1990). Ayers and Gras (1991) reported on nine years of MSA and CCN data from Cape Grim (1981 to 1989) and showed a significant seasonal (but non-linear) relationship between CCN and MSA. Ayers et al. (1991) also showed a pronounced DMS cycle with mid-summer maxima and mid-winter minima suggesting that DMS and MSA were coupled. However non-linearity of the seasonal cycles of MSA and non sea-
salt sulfate implied the existence of another source of aerosol sulfur in addition to MSA. For many years the relationship between MSA and CCN observed at Cape Grim and in other locations supported the hypothesis that DMS-derived aerosol could regulate climate by increasing CCN concentrations in response to changes in temperature or solar energy. However a review of the CLAW hypothesis, (Ayers and Cainey, 2007) suggested other sources and processes may be significant to CCN production and modulation. Gras and Keywood (2017) reported an analysis of multi-decadal CN and CCN observations from
Cape Grim focusing on relationships between the particle metrics and other variables to infer factors regulating CCN over

the multi-decadal periods. They showed that while a marine biological source of reduced sulfur appears to dominate CCN concentration over the austral summer months (December to February), other components contribute to CCN over the full annual cycle, including wind-generated coarse mode sea-salt and long range transported material.

There is also strong regional heterogeneity in the distribution of phytoplankton, with the significant latitudinal gradients, and the highest concentrations centred south of the circumpolar trough near the sea ice region in the high Southern Ocean latitude (Deppeler and Davidson, 2017). The mechanisms for the transport, chemistry and microphysics associated with the transformation of DMS emission to CCN formation are complex, and combined with this spatial heterogeneity, could impact the variability of aerosol populations in the region. Sanchez et al. (2021) recently reported on airborne aerosol measurements made during SOCRATES and showed that air masses with high CCN concentrations relative to the other regions in the Southern Ocean had always crossed the Antarctic coastline, where elevated phytoplankton emissions are known to occur. Also during SOCRATES, Twohy et al. (2021) measured aerosol types below, in and above clouds over the Southern Ocean and found biogenic sulfate and MSA made the greatest contribution to CCN. CCN and aerosol chemical composition data from CAPRICORN2 (reported in more detail in this paper) supported the observations in the airborne data reported by Twohy et al. (2021) and Sanchez et al. (2021). Alroe et al. (2020) recently presented a two-week study on the latitudinal aerosol gradients of the Southern Ocean directly south of Hobart. The data presented in this manuscript extend on the work of Alroe et al. (2020) who presented data from a short summer-time voyage that spanned all latitudes of the Southern Ocean south of Australia. In this study, voyage data from the MARCUS and CAPRICORN2 campaigns is utilised which span an entire five-month period to give a broader understanding of spatial and temporal patterns. Documenting this latitudinal gradient provides data and information that is required to evaluate the ability of models that include aerosol chemistry and pathways to simulate clouds and precipitation in this important region of the globe.

## 2 Methods

In-situ measurements were made during two research campaigns: MARCUS and CAPRICORN2. The voyage tracks of all the campaigns are shown in Figure 1.

### 2.1 Measurement Campaigns

The MARCUS (Measurements of Aerosols, Radiation and CloUds over the Southern Oceans) campaign occurred between October 2017 and March 2018 aboard RSV Aurora Australis during its summer season resupplying Australia's Antarctic research stations from its port in Hobart, Australia. In this campaign the United States Department of Energy (DOE) Atmospheric Radiation Measurement (ARM) Program Mobile Facility 2 (AMF2) Aerosol Observing System (AOS) (https://www.arm.gov/capabilities/instruments/aos) was deployed aboard the Aurora Australis to collect data in sea-ice locations inaccessible to platforms without ice-breaking capability. Because the campaign was supplementary to the resupply operations of the Aurora Australis, the Mobile Facility could only be mounted on the monkey island, slightly to the fore of the smokestack. The chosen location on the ship, while ideal for some measurements, was positioned directly adjacent to the ship's

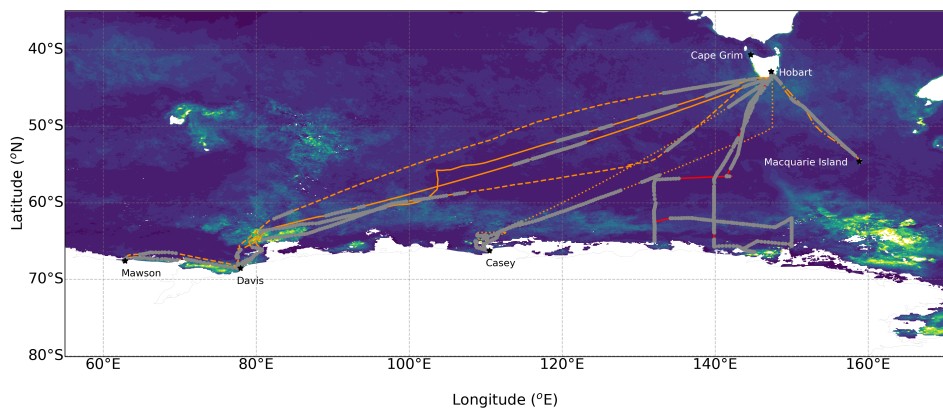

**Figure 1.** The voyages tracks from MARCUS (various orange tracks) and CAPRICORN2 (red track) overlaid on MODIS-Aqua Chl-a concentrations averaged over the measurement period from Nov 2017 to March 2018. Grey markers show voyage locations where exhaust-free data were obtained.

exhaust (see Figure A1), which when combined with the operations of the ship during the campaign, resulted in almost 90% of data being contaminated with the platform's own exhaust, limiting the usable data (detailed in Section 2.3, below).

120  Fortunately, simultaneous measurements in a similar geographic region occurred as part of the CAPRICORN2 (Clouds, Aerosols, Precipitation, Radiation, and atmospheric Composition Over the southeRn oceaN) campaign aboard the RV Investigator during January - February of 2018. While these measurements were also affected by RV Investigator's own exhaust, this was limited to less than 15% of data due to the ship operations predominantly requiring orientation into the wind, and the location of the air sampling inlet being as far fore on the ship as possible, well away from the exhaust (Figure A1). Aerosol

125  measurements during this campaign were made in the aerosol laboratory directly below the air sampling inlet on the RV Investigator, and form part of the permanent observations aboard the vessel.

Both platforms host standard meteorological stations, deployed in duplicate, which run as part of the respective ongoing underway systems and whose data are used as part of the analyses in this manuscript. Underway data, together with their associated metadata, are publicly available (Facility, 2018; Symons, 2019a, b, c, d).

## 2.2 Aerosol measurements

### 2.2.1 Cloud Condensation Nuclei

Number concentrations of CCN were measured continuously at 1 Hz at a range of supersaturations on both platforms using a commercially available continuous-flow, streamwise thermal-gradient CCN counter (CCNC, model CCN-100, Droplet Measurement Technologies, Longmont, CO, USA).

During the MARCUS campaign, the instrument was housed within the ARM Mobile Facility. The instrument was configured to sequentially measure at a range of supersaturations in sequence, including (in order): 0.0, 0.1, 0.2, 0.5, 0.8 and 1.0% supersaturation. This pattern was repeated hourly, resulting in 10 minutes of data at each supersaturation, the first three of which are removed to allow for the stabilisation of instrument conditions. The remaining data were then quality controlled for periods when instrument parameters were out of manufacturer specification, then filtered for exhaust contamination (see Section 2.3, below), before hourly statistics being calculated for each supersaturation.

During the CAPRICORN2 campaign, the instrument was housed in the aerosol laboratory. This instrument was configured similarly to the MARCUS instrument, but with the hourly supersaturation sequence including (in order): 1.0, 0.6, 0.5, 0.4, 0.3, 0.2% supersaturation. Similar quality control procedures were undertaken for this instrument as for the MARCUS data, however because this instrument was calibrated at the Droplet Measurement Technologies laboratory in Colorado, pressure corrections for the supersaturations were made which resulted in the actual measured supersaturations being 0.055% higher than the set points (e.g., 0.5% was actually 0.555%). This was not the case with MARCUS data since calibrations were undertaken at sea level. Flows on both instruments were set to 0.5 $Lmin^{-1}$ and checked regularly to ensure they remained within specification.

Throughout this study, the two supersaturations common to both campaigns were utilised in our data analysis - 0.2 and 0.5% (referred to throughout this manuscript as $CCN_{0.2}$ and $CCN_{0.5}$, respectively). It is noted however that recent aircraft measurements in the region suggests that 0.3% is likely to be the best representation of actual environmental conditions in the region (Fossum et al., 2018; Sanchez et al., 2021; Twohy et al., 2021).

The full metadata record and measurement data for the MARCUS campaign are available at Kulkarni et al. (2018), however these data have not been filtered for exhaust contamination. An exhaust filtered and reprocessed dataset was undertaken specifically for this study and data are available at (Humphries, 2020). Fully processed and exhaust filtered data for CAPRICORN2 are available at (Humphries et al., 2021a).

### 2.2.2 Condensation Nuclei

Number concentrations of condensation nuclei (aerosols) larger than 10 nm ($CN_{10}$) were measured continuously at 1 Hz on both platforms using condensation particle counters (CPC Model 3772, TSI Inc. Shoreview, MN, USA). The CPC draws sample air continuously through a chamber of supersaturated 1-butanol, which condenses and grows particles to super-micron sizes where they are counted individually by a simple optical particle counter. For this study, the manufacturers default 50% counting efficiency (D-50) was used, and is defined at 10 nm. The sample flow rate is typically regulated by an internal critical orifice

(MARCUS instrument was configured this way). However the critical orifice in the CAPRICORN2 instrument was replaced with a mass flow controller (MFC, Alicat Scientific Model MC 5SLPM) to ensure more accurate flow control, particularly in a marine environment, where the critical orifice can becomes quickly blocked with sea-salt aerosol. The MFC was calibrated using an external low-pressure drop flowmeter (Sensidyne Gilibrator, St. Petersburg, FL, USA). Flows on both instruments were set to 1.0 $L.min^{-1}$ and checked regularly to maintain specification. Data are filtered for periods of instrument zeros, flow checks and other outages, as well as for platform exhaust, before hourly statistics were calculated.

As with CCN data, MARCUS ($CN_{10}$) data found at the ARM data repository (Kuang et al., 2018) are not filtered for exhaust contamination. Exhaust filtered and reprocessed data from both MARCUS and CAPRICORN2 can be found at Humphries (2020) and Humphries et al. (2021a) respectively.

### 2.2.3  Aerosol composition

The composition of non-refractory aerosol was measured continuously during CAPRICORN2 using a Time-of-Flight Aerosol Chemical Speciation Monitor (Aerodyne ToF-ACSM, Billerica, MA, USA Fröhlich et al. (2013)). The ACSM's aerodynamic lens inlet permits particles with diameters between 70 and 700 nm (Liu et al., 2007) to enter the vacuum chamber before impacting onto a vaporiser heated to $600^{o}C$ where non-refractory particles are vaporised and then ionised with electron impact ionisation. Ions are then directed into a time-of-flight mass spectrometer (0-400 amu) resulting in 1 Hz mass spectra. Aerosol spectra are identified above background air by continuously switching between particle-free (through a HEPA filter) and sample air every 20 seconds. Aerosol spectra are used to calculate 10 min averages of sulfate, nitrate, ammonium, chlorine, methanesulfonic acid (MSA) and a grouped 'organics' class. It is important to note that because of the size selection and the refractory nature of sea-salt, the actual concentrations reported for chlorine have yet to be calibrated to obtain a correction factor and values are only used in a relative manner in this manuscript. Ammonium nitrate and sulfate calibrations were run prior to and after the voyage, as is standard operating procedure.

During CAPRICORN2, time integrated aerosol composition measurements were made alongside the on-line ACSM measurements using a PM1 size selective inlet (BGI SCC model 2.229, Butler, NJ, USA) and 47 mm quartz filters. Each filter sampled for 1-2 days (20-48 hours) at a flow rate of 16.67 vLPM (required for the SSC) controlled by a MFC (Alicat Scientific Model MC 20SLPM). To prevent the filters being contaminated (and overwhelmed) by exhaust aerosol, the system was placed on a switching controller which ceased sampling when relative winds directions were between $90^{o}$ and $270^{o}$ and CN concentrations were above a threshold value. This meant the PM1 sampling system was switched on and off throughout the sampling period so that total volumes through each filter ranged between 14 to 26 $m^3$. The instantaneous volumetric flow rate from the MFC was recorded and totaled by a electronic flow totalizer (Amalgamated Instruments Co., model PM4-IVT-DC-8E, Hornsby, NSW, Australia).

Five field blanks were collected approximately weekly throughout the campaign. Field blanks involved carrying out the full filter change process with the sample pumps remaining switched off. After sampling, filters were enclosed in clean aluminium foil and frozen until they could be analysed post-voyage. The soluble ion concentrations were determined using high performance anion-exchange chromatography with pulsed amperometric detection (HPAEC-PAD) was measured at the CSIRO

laboratories in Aspendale, Victoria. The filters were be extracted in 10 ml of 18.2 mΩ de-ionized water. The sample was then preserved using 1% chloroform. Anion and cation concentrations are determined with a Dionex ICS-3000 reagent free ion chromatograph. Anions are separated using a Dionex AS17c analytical column (2 x 250 mm), an ASRS-300 suppressor and a gradient eluent of 0.75 mM to 35 mM potassium hydroxide. Cations are separated using a Dionex CS12a column (2 x 250 mm), a CSRS-300 suppressor and an isocratic eluent of 20 mM methanesulfonic acid. All values reported in this manuscript are blank corrected.

Aerosol composition data measured by the Tof-ACSM and on the PM1 filters during CAPRICORN2 are available at Humphries et al. (2021a).

## 2.3 Platform exhaust

Removal of exhaust contaminated data is a critical step required before using any aerosol composition data from diesel-powered ship platforms. For aerosol data, exhaust signals are typically orders of magnitude higher than ambient data, given the strength and proximity of the source to measurement points. Other sources of contaminated air, for example the incinerator and indoor air vents, are minor in comparison with the exhaust, but are captured through the filtering process described below, largely because these vents are colocated with engine exhaust emissions and have a similar effect on measurements, albeit smaller. The engine age, fuel type, ship architecture (e.g., how the air flows around the ship and creates local eddies), relative locations of exhaust and air sampling inlet as well as the operations of the ship during measurements, also affect how much impact the exhaust has on the atmospheric measurements. MARCUS was undertaken aboard the Aurora Australis, an ice-breaker commissioned in 1989, powered by two Martsila medium-speed diesel engines (one 16V32D and one 12V32D, producing a total of 10 000 kW) and burning standard marine grade fuel oil. As shown in Figure A1, the ARM measurement container was located directly adjacent to the exhaust pipe of the ship, meaning that a large proportion of wind conditions were able to push exhaust into the sampling inlet. The MARCUS campaign was also supplementary to the usual resupply voyages of the Australian Antarctic Program, and consequently, the direction of the ship was rarely optimal for atmospheric measurements - instead being more focused on swell and sea-ice conditions.

In comparison, the CAPRICORN2 voyage was undertaken aboard the RV Investigator, which was purpose built for marine science, and specifically incorporated atmospheric measurements into its design, resulting in an architecture optimised for minimal exhaust impact. The atmospheric sampling inlet is located as far forward on the vessel as possible (see Figure A1), resulting in a significant distance from the exhaust. The ship itself is powered by three nine-cylinder MaK diesel engines coupled to a 690V AC generator, and burns automotive-grade diesel fuel (as opposed to residual (heavy) fuel oil used by most vessels). During the CAPRICORN2 voyage, the ship was largely positioned to face directly into the wind, ideal for marine Conductivity, Temperature and Depth (CTD) measurements occurring during the voyage, and where possible, transits between marine targets were optimised for favourable wind conditions to maximise the collection of exhaust-free atmospheric data.

On both platforms, wind direction was found to be a poor parameter for filtering exhaust (Humphries et al., 2019), likely because of the eddies that form around the ship's superstructure and create differences between wind directions measured by meteorological instruments and those experienced at measurement height. Instead, we used differences in composition between

the exhaust and clean background air to identify and remove exhaust influence. The exhaust was identified the same way on both platforms. A first pass was undertaken using the automated exhaust identification algorithm described in (Humphries et al., 2019). This algorithm was designed to strike a balance between accurately removing obvious exhaust signals, but not being too over-zealous and unwittingly removing clean data. This balance results in the correct removal of about 95% of exhaust signals. Manual filtering is undertaken as the next step, and identifies any rapid increases in CN, black carbon, carbon monoxide or carbon dioxide concentrations, and in the case of the Aurora Australis, any drops in ozone resulting from titration from engine-produced nitrogen oxides. It is noted here that the CN data is relied on most heavily during this process, because of its high time resolution and highest sensitivity to the exhaust signal, which typically changes by orders of magnitude relative to background data.

Because of the differences between the platforms and ship operations, the resulting proportion of clean data available for each campaign was significantly different. For MARCUS, only 11% of data was exhaust-free, resulting in approximately 500 hourly data points for CCN over the 129 days at sea. In contrast, over 86% of data during CAPRICORNII was exhaust free, resulting in over 760 hourly data points from the 42 day campaign. An example time series of MARCUS CCN data and the amount of data removed by exhaust filtering is shown in Figure A2. Overlayed on Figure 1 ship tracks are the locations where exhaust-free data exists for the campaigns. Despite the significant data loss associated with exhaust contamination, the latitudinal coverage of the data is reasonable, with each of the $5^o$ latitudinal bins having over 110 data points in each, with the bins south of $60^o$S containing over 450 data points (as shown in Figure 2).

## 2.4   Trajectory analyses

The HYSPLIT (HYbrid Single-Particle Lagrangian Integrated Trajectory) model (Draxler and Hess, 1998) was used to calculate air parcel trajectories. In this study, HYSPLIT was used to calculate back trajectories in order to evaluate source regional and vertical source locations for the various categories. Trajectories were calculated using the Global Data Assimilation System (GDAS) reanalysis (Kanamitsu, 1989). The model was setup to utilise 1 degree horizontal resolution reanalysis data and vertical motion utilised the model vertical velocity method. Calculations utilised surface invariant geopotential, surface 10 m horizontal (U and V) winds, 2 m surface temperature, and U, V, W (vertical wind), temperature and humidity on pressure levels from 1000 to 20 hPa. Each trajectory calculation provided hourly three-dimensional air parcel locations for a total time-span of up to five days in order to limit uncertainty magnification. Trajectories were initiated at the ship's location for every hour of the cruise at heights of 10 m and represent heights above ground level.

Once calculated, trajectories were divided into $5^o$ latitudinal bins based on their starting location. Only trajectories corresponding to exhaust free aerosol data were used. For each set of latitudinally-binned trajectories, frequency plots were calculated by summing the number of times trajectories passed through a map, binned such that the horizontal resolution of the boxes was $0.5^o$ and with linearly spaced, 10 m vertical bins. Resulting plots were smoothed using the kernel-density estimations using Gaussian kernels (implemented with Python SciPy's guassian_kde function). The bandwidth was determined by taking the average effective samples in each bin calculated using Scott's factor (Scott, 2015).

Total precipitation along each trajectory was calculated using ERA-5 reanalysis data (ECMWF (2018), variable name "tp",
hourly time steps with spatial grids of 0.25 degrees in both latitude and longitude) . For each step in each trajectory, the
precipitation at the time and location was retrieved from the reanalysis data, then the values for each step were summed,
resulting in a single total precipitation value for each trajectory. As before, only trajectories corresponding to exhaust-free
measurement periods were included in the analysis.

## 3   Results

Despite the significant removal of data due to exhaust in the MARCUS campaign, the utilisation of the CAPRICORN2 campaign data, which occurred at the same time in the same regional area, meant that division of the data into $5^o$ latitudinal bins
resulted in sample numbers high enough to calculate robust statistics in each bin, while having reasonable latitudinal resolution.
In Figure 2, violin plots show measurements distribution in each latitudinal bin for each of $CCN_{0.2}$, $CCN_{0.5}$ and $CN_{10}$ (full
statistics for each bin are presented in Table A1 and latitudinal gradients for each voyage for $CCN_{0.5}$ and $CN_{10}$ are presented
in Figure A3 and Figure A5, respectively). The highest concentrations (means of 169, 322 and 681 $cm^{-3}$ for $CCN_{0.2}$, $CCN_{0.5}$
and $CN_{10}$, respectively) for all parameters are unsurprisingly observed in the northern-most bin, 40-45$^o$S which is closest to
the coast of Tasmania, Australia, resulting in increased continental and anthropogenic influence. Moving south, $CN_{10}$ concentrations appear to be stable (300-400 $cm^{-3}$) from $45^oS$ to $65^oS$. This isn't the case with CCN at both supersaturations,
which show slightly elevated concentrations in the 45-50$^oS$ bin (131 and 197 $cm^{-3}$ for $CCN_{0.2}$ and $CCN_{0.5}$, respectively),
after-which it becomes reasonably constant from $50^oS$ to $65^oS$ ($\tilde{1}$00 and 150 $cm^{-3}$, respectively).

Measurements taken from nearby land-based research stations at Macquarie Island (54$^o$30'S, 158$^o$57'E; Humphries (2020))
and Cape Grim (40$^o$39'S, 144$^o$44'E; Gras and Keywood (2017)) were utilised to compare with the ship-based measurements
considered here. Macquarie Island is one of the research stations visited as part of the seasonal resupply operations undertaken
by the Aurora Australis during MARCUS (voyage 4 of the 2017/18 ship schedule) and consequently, data from MARCUS and
from the land-station are directly comparable. In addition, the upwind fetch of Macquarie Island is dominated solely by the
Southern Ocean, so other voyage data at these latitudes, which happen to all be within the upwind fetch of Macquarie Island,
are also comparable. The Cape Grim station is classified as a Global station in the Global Atmospheric Watch station, being
representative of a globally significant region, so although the location is a reasonable distance from the ship measurements, we
use them here with confidence. In Figure 2, the median values from November 2017 to March 2018 (chosen to coincide with
the MARCUS campaign period) are presented from both stations for each available parameter, and overlaid on the latitudinal
gradients in their respective latitudinal bins. Two data sets are presented for Cape Grim: all data (red) and baseline data
(green). Baseline refers to periods that represent a clean marine background with fetches from the Southern Ocean with little
to-no continental influence (Gras and Keywood, 2017).

For both CCN and CN data, values measured at Macquarie Island agree well with those measured aboard the ships, and
are likely to be a good representation of Southern Ocean mid-latitudes. Cape Grim values are markedly higher for $CN_{10}$ and
non-baseline selected CCN data, and generally agree well with ship data in the respective latitudinal bin. Cape Grim baseline

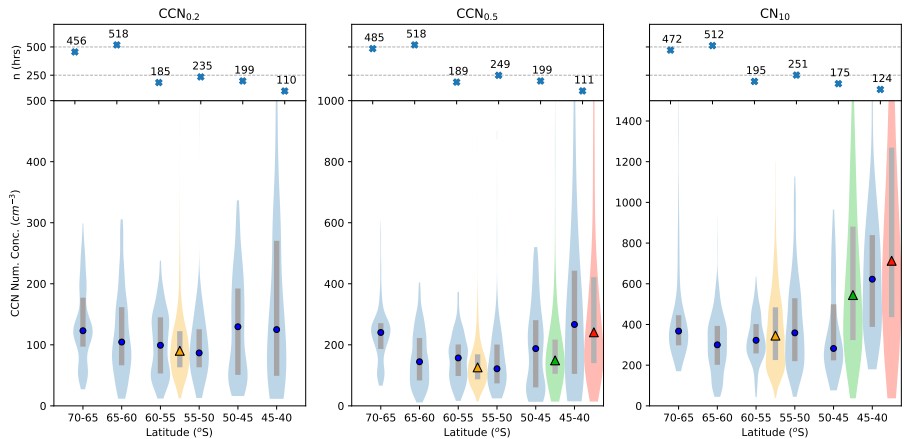

**Figure 2.** Latitudinal distributions of CCN (at 0.2% and 0.5% supersaturation, left and middle respectively) and $CN_{10}$ (right) from collated data from both MARCUS and CAPRICORN2 campaigns shown in blue. Data are binned into $5^o$ latitudinal bins and plotted as violin plots with medians (circles) and 25th and 75th percentiles (grey box) shown. Data from Nov 2017 to Mar 2018 from Macquarie Island ($54^o30$'S, $158^o57$'E; orange) and Cape Grim ($40^o39$'S, $144^o44$'E; left, green is baseline (Rn $< 100$ mBq, wind directions between $190^o$ and $280^o$); red is all data) at their respective latitudinal bins. Subplots above each plot show the number of hourly data points in each bin. Note the y-axes scales are custom for each data set.

data appear to be lower than ship-based measurements in their respective latitudinal bin, which may be explained by ship data not being filtered for baseline criteria, and are consequently likely to include some level of continental/anthropogenic influence. Non-baseline $CN_{10}$ data from Cape Grim are higher than those measured on the ship, and this is likely because of
300    the influence of fine-mode aerosol emissions from the metropolitan region of Melbourne, as well as emissions from Tasmania, both of which can influence Cape Grim measurements in non-baseline conditions. Curiously, ship-based CCN data are similar to non-baseline Cape Grim data, and significantly higher than baseline data (which is actually similar to the higher latitude bins), which suggests ship-based measurements were influenced significantly by continental sources while measuring at these latitudes, a result confirmed by trajectory analyses presented later in the manuscript.
305    Most striking in the latitudinal distribution is the statistically significant increase in all aerosol parameters in the southernmost bin along the Antarctic coastline (p $< .001$ compared to 60-65$^o$S bin). While most pronounced in $CCN_{0.5}$ (mean concentrations increase by 50% compared to mid-latitudes with 30% increases for both $CCN_{0.2}$ and $CN_{10}$), corresponding changes are also apparent in the CCN/CN ratio, wind speed, and more significantly in precipitation, as shown in Figure A7. It is likely that the larger increase in $CCN_{0.5}$ relative to $CCN_{0.2}$ in this bin is a result of the changing size distributions and aerosol compo-
310    sition when moving between air-masses. To explore this apparent change in composition further, we examined more closely the CAPRICORN2 data which provided both real-time, and filter-based aerosol composition data. Latitudinal aerosol composition data from this voyage is shown in Figure 3 alongside binned wind speed and precipitation data.

While the increase in this southern-most bin was not significant in the MARCUS $CCN_{0.2}$ data, CAPRICORN2 data (Figures 3, A3, A4, A5, A6) show significant increases in CCN at all supersaturations, in the CCN/CN ratios, as well as changes in the dominant species contributing to the aerosol composition (Figure 3). Andreae (2009) found that the CCN0.4/CN ratios from a wide range of environments averaged 0.4, agreeing well with mid-latitude data from this study, but highlighting the unique nature of the polar populations. Aerosol composition further south is dominated by sulfur-based particles, consistent with the established literature (e.g. Fossum et al., 2018; Schmale et al., 2019), whose relative contribution to aerosol mass (as measured by the ToF-ACSM) was around 60-70% at lower-latitudes, but increased to its maximum in the southern-most bin, reaching over 80%. We note here that while trends from the filters are similar to those observed on from the ToF-ACSM, the differences observed as likely the result of different sampling techniques: the ToF-ACSM measuring non-refractory aerosol composition, while the filters are analysed for soluble ions. The increases in CCN ratio could be driven by a stronger source of sulfur precursors (sulfate and MSA derived from DMS) emitted from enhanced phytoplankton near the Antarctic continent (Figure 1), but are likely to also be driven by a significant drop in precipitation which would preferentially scavenge CCN compared to other aerosols. Chloride, which is used as a proxy for sea spray aerosol, is observed to be dominant at lower latitudes (and varies proportionately to wind speed, Figure A9) but reaches its minimum in the high latitude bin. This significant reduction in the high-latitude bin is consistent with the combined effect of decreased wind speeds and the occurrence of sea-ice covering the ocean surface, resulting in a substantially lower source strength which outweighs the reduced precipitation sink. Interestingly, comparison of the distributions of CCN with the sulfur and chloride composition measurements suggests that while sea-salt aerosol contributes an important baseline to CCN numbers, the variability, and in particular the vast population of CCN at high latitudes, is driven by sulfur-based aerosols, similar to what has been reported by Vallina et al. (2006) and others. Recent work by Fossum et al. (2020) suggest an inverse relationship between sea-salt aerosol concentrations and sulfate CCN activation, which could also help to explain this change in the southern-most bin.

To further understand the source regions of the observed latitudinal changes, we calculated the trajectories for each hour of exhaust-free data during the five voyages. In Figure 4, these trajectories, split into the six latitudinal bins based on each trajectory's end location, are shown as density plots. As expected, the northern-most bin shows influence primarily from the marine boundary layer upwind of the measurements, and significant influence from both Tasmania and the more heavily populated areas of south-eastern Australia. This trajectory footprint is consistent with $CN_{10}$ ship values in this bin being higher than baseline values at Cape Grim (Figure 2), where these continental and anthropogenic influences are excluded. Note that non-baseline values from Cape Grim are higher than those from the ship-measurements in this bin, despite similar source regions. This is likely to be explained simply by the closer proximity of Cape Grim to these anthropogenic sources than the ship measurements. For bins from $45^{o}S$ to $60^{o}S$, all fetches reside within the Southern Ocean's marine boundary layer upwind of the measurements, with only a small subset of trajectories arriving from the free-troposphere. Unlike other bins, the southern-most bin has fetches that consist primarily of coastal and Antarctic continental latitudes, with minimal marine influence. The altitude plot shows the boundary layer influence is significantly reduced, and instead, fetches are distributed across a range of heights in the troposphere. Interestingly, the $60^{o}S$ to $65^{o}S$ bin is a mix of marine boundary layer and free tropospheric fetches, and is likely a result of the atmospheric polar front varying at latitudes covered by this bin.

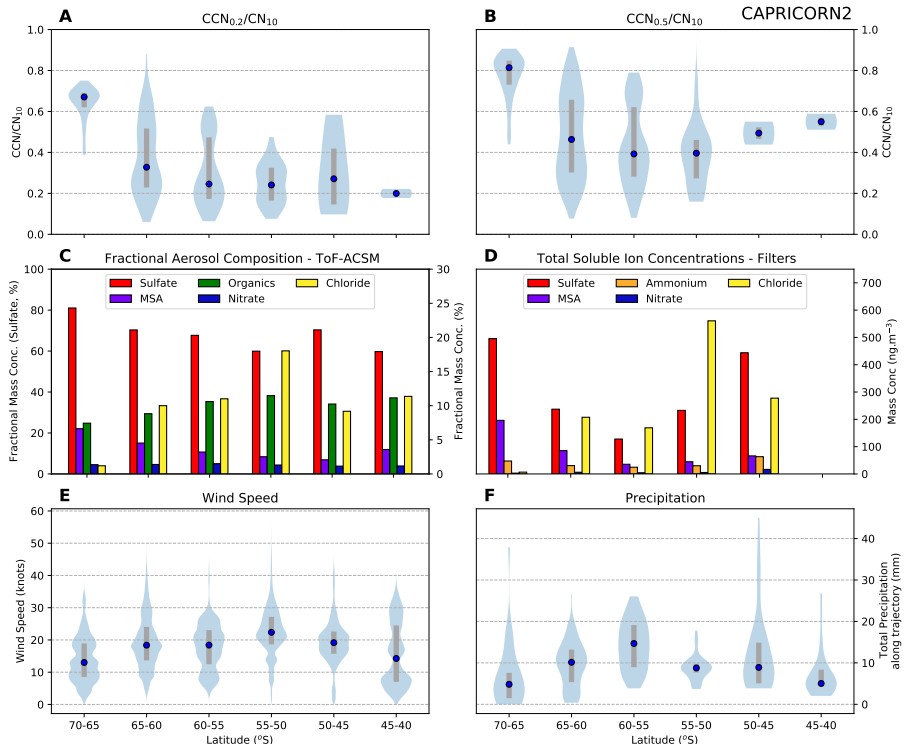

**Figure 3.** Latitudinal distributions for parameters measured during CAPRICORN2 highlighting the aerosol composition and source and sink mechanisms. The $CCN/CN_{10}$ ratios for 0.2% and 0.5% supersaturation are shown in A) and B), respectively; the major aerosol chemical components from the ToF-ACSM in C); the total soluble ions from PM1 filters, shown in D); E) shows the wind speed measured onboard the vessel; and F) shows the total precipitation calculated using ERA5 reanalysis data along the backward trajectory for each measurement. Note that on plot C, the sulfate is split to the left axis to enable better visibility of trends of other components.

The trajectory analysis shows the air-mass histories in the region are consistent with more detailed trajectory studies undertaken previously (Humphries et al., 2016; Alroe et al., 2020). In particular, measurements from the mid-latitudes of the Southern Ocean are found to have air mass histories confined primarily to the marine boundary layer, whereas the closer the approach to the East Antarctic continent, the greater the influence from the free troposphere of the Polar Cell. These large-scale air-flow differences are likely the leading cause of the differences between the mid and high-latitude aerosol properties measured in this study. Given the remoteness of the region and the limited number of aerosols sources, typically phytoplankton emissions and sea spray, it is likely that the aerosol sources are consistent across all latitudes. However differences observed in the atmosphere are driven by two factors: 1) air-mass fetches altering the efficiency and strength of the sea-salt source, and 2) sources of secondary aerosols are the same, but because of the differences of air-mass transport, the properties of the aerosol populations differ. For example, if high latitude transport pathways take the bulk of phytoplankton emissions from the sea-ice region, into the Antarctic free troposphere before being brought back to the surface (as proposed by Humphries et al., 2016)),

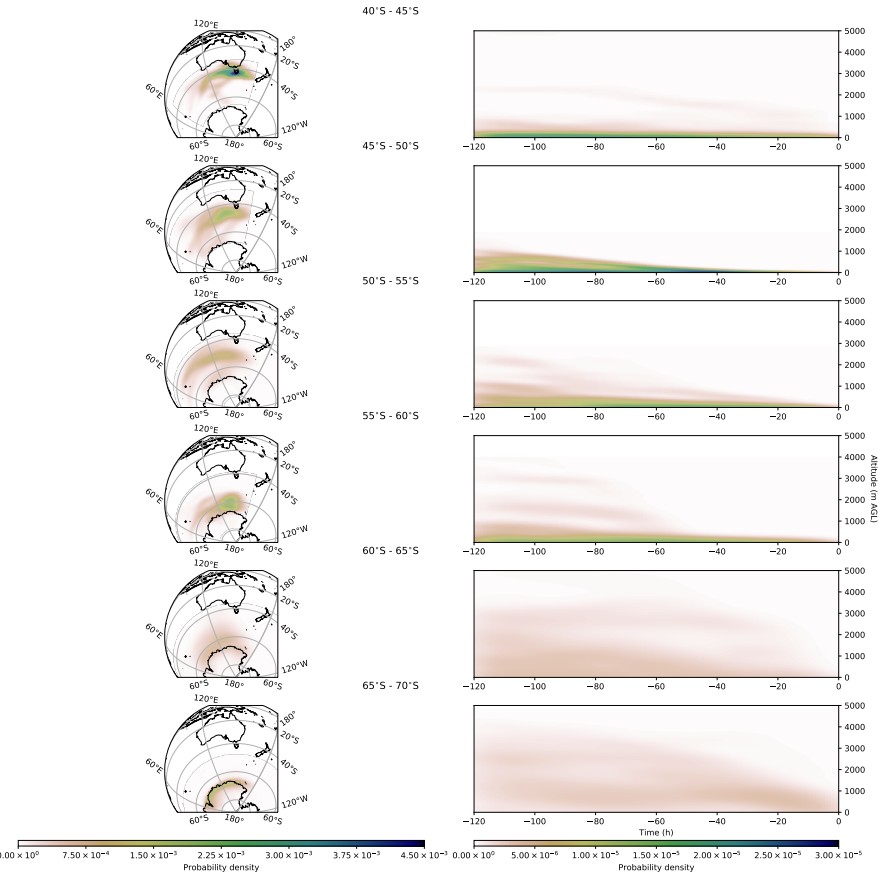

**Figure 4.** Trajectory frequency plots showing the spatial footprint of measurements in each latitudinal bin in each dimension. Five day backward trajectories were calculated for each hour of valid, exhaust-free data, and then the numbers of times trajectories passed through $0.5^o$ bins was summed before smoothing, resulting in frequency plots. Map plots are shown in the left column, with the starting locations of each of the trajectories used for each binned plot shown in black. The right column shows the frequency plots as a function of time and altitude, giving an indication of the dominant atmospheric layers important in each region.

the increased precursor concentrations may result in more aerosol nucleation and growth in the free troposphere, resulting
360   in the enhanced CCN concentrations observed in the high latitude bin. In addition, free tropospheric aerosols would be less exposed to the high surface area of sea-salt aerosol that typically dominate the aerosol mass in the marine boundary layer, and so be less likely to be scavenged, resulting in the higher CCN number concentrations.

## 4 Discussion

These data suggest that there are three distinct latitudinal regions that govern the Southern Ocean aerosol populations in the summer season: the northern sector north of $45^oS$; the mid-latitude sector ($45$-$65^oS$); and the high latitude coastal region of Antarctica ($65$-$70^oS$).

Unsurprisingly, the northern sector exhibits the most anthropogenic and continental influence, resulting in aerosol concentrations approximately twice that of those in the open Southern Ocean for both $CN_{10}$ and $CCN_{0.5}$. For $CCN_{0.2}$ this distinction is not so clear, suggesting that the same source as mid-latitudes is driving CCN concentrations at this supersaturation (presumably sea-salt). This would suggest that aerosols arising from anthropogenic and continental sources are less hygroscopic than sea salt, which is consistent with what is expected from the literature (e.g. Swietlicki et al., 2008).

The mid-latitude observations are consistent throughout a large range of latitudes, being dominated by sea-salt and sulfur-based aerosols (Figure A9). Given the lack of any land-masses in this region, the primary aerosol sources are driven by wind-produced sea-salt, and secondary aerosol formation typically resulting from both local and long-range transport of aerosol precursors emitted from biological sources, chiefly DMS from phytoplankton. The dependence of sea-salt aerosol concentrations on wind speed and precipitation is striking, being directly and inversely proportional, respectively. This relationship breaks down in the high latitude bins where sea-ice cover impacts on the wind mechanism of sea-salt aerosol production.

While observations in both the northern and mid latitudes have been noted previously in the literature, the high-latitude observations are novel. This is in part driven by how remote the region is, and how infrequently it has been sampled in the past (e.g., high latitude Southern Ocean measurements off East Antarctica are reported only by Humphries et al. (2016); Alroe et al. (2020); Simmons et al. (2020); Schmale et al. (2019)). For example, the recent SOCRATES aircraft campaign (McFarquhar et al., 2021) involved measurements made on flights originating from Hobart. However because of the limited range of the aircraft, measurements could only be made to $62^oS$, so that the significant change in aerosol populations at higher latitudes could not be observed.

Measurements have been made in other parts of the Antarctic sea ice (e.g. Davison et al., 1996; Fossum et al., 2018; Schmale et al., 2019). Typically these sectors do not show the step changes observed in the East Antarctic sea ice measurements, and instead are reasonably well represented by continental measurements (e.g. Asmi et al., 2010; Hansen et al., 2009; Hara et al., 2011; Ito, 1993; Järvinen et al., 2013; Koponen et al., 2003; Pant et al., 2011; Samson et al., 1990; Virkkula et al., 2009; Weller et al., 2011; Hara et al., 2020)). During a campaign around the Antarctic Peninsula in summer 2015, Fossum et al. (2018) observed two distinct air-masses: those coming from continental Antarctica and those from the marine region north of the polar front. They found that, despite the differing composition of the two air-masses which reflected observations described in this manuscript (i.e. a decrease in sea-salt in air-masses from the south), CCN concentrations at realistic supersaturations for this region (0.3%) remained relatively constant at around 200 $cm^{-3}$. Schmale et al. (2019) report CCN concentrations determined while circumnavigating Antarctica between December 2016 and March 2017, with leg 2 of the voyage passing closest to the Antarctic continent (mainly west Antarctica). Median CCN at 0.02% SS during Leg 2 was 111 $cm^{-3}$, similar to the concentrations measured south of $65^oS$ in this study, and the fraction of CN acting as CCN was highest near the Antarctic

continent, also observed in our study. Schmale et al. (2019) suggested this could be due to differences in aerosol chemical composition, with a combination of many cloud processing cycles and greater supply of DMS oxidation products resulting in particles large enough to act as CCN.

Contrary to the West Antarctic region, the region of the East Antarctic coast included in this study is not well represented by measurements on the Antarctic continent itself, a phenomenon driven by well defined air-mass transport which isolates the continent from the sea ice region (Humphries et al., 2016). This result was true for springtime measurements, and further work by the same authors (currently unpublished), suggests that the meteorology that leads to this phenomenon may break down both during summertime and around the Antarctic Peninsula. Since the majority of measurements in the region occur in

summer and at either Antarctic stations or at lower latitudes, the East Antarctic coastal region remains one of the more poorly represented regions of the world.

    The change in aerosol properties at this high-latitude bin is consistent with the crossing of the Antarctic atmospheric polar front, as first described by Humphries et al. (2016) and observed by both Alroe et al. (2020) and Simmons et al. (2020). While values observed in this manuscript are in line with those previously observed across the polar front (Alroe et al., 2020; Simmons

et al., 2020), this data set adds confidence that the change observed in these previous studies is an enduring phenomenon across a wider range of East Antarctic longitudes and seasons. While the definition of the Antarctic polar cell is traditionally understood in terms of climatological averages, it is evident that a very real-time boundary exists that can be seen in the atmospheric composition observations, which isn't necessarily observed in the meteorological variables from which it is typically defined (i.e. a sharp change in wind direction and air temperature). Because of this, and to create a distinction from the traditionally

defined meteorological front, we introduce a new term to define it here as the Atmospheric Compositional Front of Antarctica (ACFA), which represents the northern boundary of the region that extends south to approximately the Antarctic coastline - a region we term the Antarctic Sea Ice Atmospheric Compositional Zone (ASIACZ). It is important to note that while the aerosol properties in the ASIACZ are not captured by surface measurements on the Antarctic continent, nor those in the Southern Ocean mid-latitudes, trajectory studies suggest that airmasses from this region travel both north and south, typically

above the boundary layer (Humphries et al., 2016), making this an important region of exporting aerosols and precursors from a highly biologically productive region to other regions. This could help reconcile the predicted missing aerosol source in the wider region.

    The ACFA is known to vary in time and space, and can be advected by the synoptic-scale meteorology. This is evident from the individual voyage latitudinal plots (Figures A3 and A5) where the increases in the southernmost bin latitudes differ

depending on the voyage and location. This movement can even occur within a single voyage, as evidenced clearly from the CAPRICORN2 voyage data (Figures A3 and A8), where the increase in CCN occurred at approximately $64^oS$ during one crossing at $150^oE$, and $62.5^oS$ during the $140^oE$ transect. During this voyage, we also tried to intentionally cross the front while sampling south along the $132^oE$ meridian. However we were unable to locate the front even when when travelling further south ($>65^oS$) of the ACFA's latitude just days before.

By investigating latitudinal gradients across the parts of the Southern Ocean, this work raises some important objectives for future work. A significant motivation for this work is to better inform and reconcile the radiation biases arising from poor

representation of clouds in climate and earth system models. Hence relating these observations to recent cloud observations is important, and this work is well underway. Mace et al. (2021) analyzed MARCUS and CAPRICORN2 data and found gradients in cloud droplet number concentrations in reasonable agreement with the gradients in CCN concentrations identified

here. Interestingly, Mace et al. (2021) observed a bimodal distribution in cloud properties poleward of $62.5^oS$. While one mode displayed properties of marine clouds from farther north, the second showed relatively high cloud droplet numbers and low effective radii. The bimodality was inferred to be associated with changes in air mass properties (such as CN, CCN and aerosol chemistry), identified in previous work (Humphries et al., 2016), in case study events described in Mace et al. (2021) and in further analysis of CAPRICORN 2 data currently underway. In particular these changes in air mass properties included

those identified systematically in this work (i.e. high CCN and CCN/CN and high sulfate and MSA indicative of biogenic aerosol sources) in the southern sector. The potential sensitivity of the cloud properties to these biogenic aerosol sources suggest a strong feedback with biological and photochemical activity in the region, an issue that warrants further and extensive investigation.

Further studies should also address the transition across the ACFA. A more detailed assessment of the datasets used in this

manuscript is currently underway that focuses on the transition and the chemical and physical changes that occur in gases, aerosols and clouds in this region, and will be described in a future publication. The comprehensive, continuous measurements aboard the RV Investigator also provide a perfect opportunity for understanding transition as the vessel frequently undertakes voyages into this region - albeit primarily limited to the summertime.

These conclusions are all based on the sector of the Southern Ocean between Australia and East Antarctic, and only valid

for late spring and summer measurements. It is possible that different patterns may be observed in other sectors of the Southern Ocean that are influenced by disparate continental influences, such as those around Africa and South America. Hence, it is important that future observation campaigns investigate these regions. While important circumpolar measurements such as those undertaken by Schmale et al. (2019) give an insight into this variability, campaign measurements are limited in their duration, and generally undertaken during the summertime (e.g. Humphries et al., 2016; Fossum et al., 2018). To avoid biases

that may arise by applying these conclusions to other seasons, long-term measurements, such as those undertaken at Cape Grim, Cape Point South Africa (Labuschagne et al., 2018) and the RV Investigator are needed across other longitudes of the Southern Ocean. These include remote sites such as Macquarie Island (measurements included here were limited to just two years) and research platforms that frequent the ASIACZ, such as the soon-to-be commissioned RVS Nuyina. Long term year round atmospheric aerosol data sets reveal important seasonal and annual variability, and the processes that contribute to this

variability. Hence these data will be critical for ensuring reduced biases in modelling efforts.

## 5 Conclusions

In this manuscript we collated data from two intensive research campaigns, spanning five voyages in the Southern Ocean region between Australia and East Antarctica. Aerosol microphysical and chemical properties were assessed in terms of the latitudinal variability. Three main latitudinal regions were identified: the northern sector north of around $45^oS$, where continental and

anthropogenic sources add to the background marine atmosphere; the mid-latitudes (45-65$^o$S), where the marine boundary layer populations dominate; and the far south (65-70$^o$S), termed here as the Antarctic Sea Ice Atmospheric Compositional Zone (ASIACZ), where aerosol populations are dominated by sulfur-based species derived from free-troposphere nucleation, and seaspray aerosol is significantly reduced. Aerosol concentrations were highest in the northern and southern bins, with CCN$_{0.5}$ concentrations being approximately 70% higher than mid-latitudes concentrations of around 150 cm$^{-3}$. Simultaneous measurements from nearby land-based stations were compared with these shipboard measurements and were found to be good representations of their respective latitudes, with the long-term baseline measurements at Cape Grim being representative of CCN at all northern and mid-latitudes, and CN$_{10}$ at the northern sector. Measurements from a two year campaign at the sub-Antarctic station of Macquarie Island (54$^o$30'S, 158$^o$57'E) were found to be representative of the mid-latitudes for all species. The ASIACZ region was not represented by either of these stations, and previous work suggests that measurements at research stations on the Antarctic continent are not reflective of this spatially significant region. Further measurements are important to capture the spatial, seasonal and inter-annual variability across the different latitudes, as well as the longitudinal variability that is likely when investigating the Southern Ocean regions around Africa and South America.

*Data availability.* Datasets are available at relevant citations within the manuscript.

**Appendix A: Appendix**

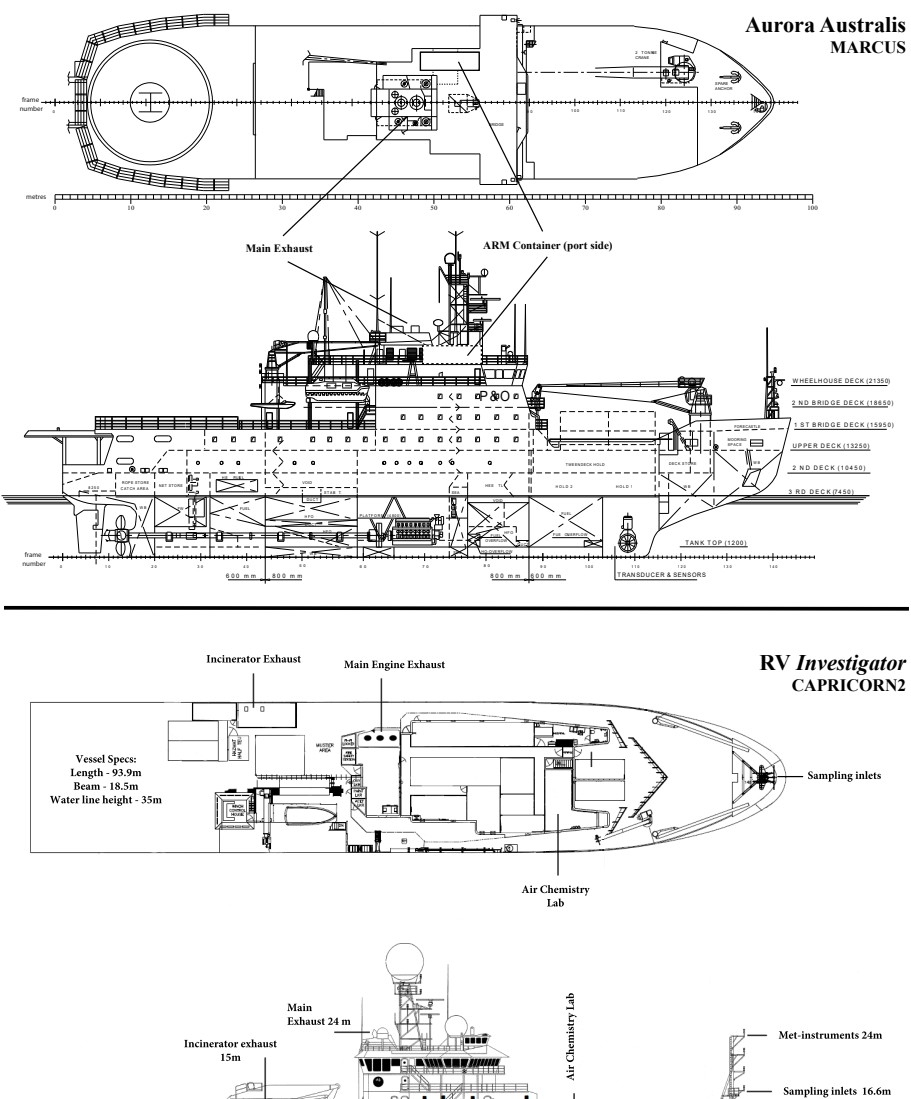

**Figure A1.** Ship schematics of both the Aurora Australis (top), used for the MARCUS campaign, and the RV *Investigator* (bottom), used for the CAPRICORN2 campaign. Contamination of samples by the ship's own exhaust is the primary driver in the removal of data, with only 11% (500 hours) and 86% (760 hours) of data remaining after exhaust filtering for MARCUS and CAPRICORN2 campaigns, respectively. Exhaust contamination is driven by the proximity of the measurements to the exhaust, the age and cleanliness of the engine, together with ship operations during voyages and whether these operations align the ship with favourable wind directions that push the exhaust away from the sampling inlet. All these factors contributed to the high contamination of MARCUS data relative to CAPRICORN2 data.

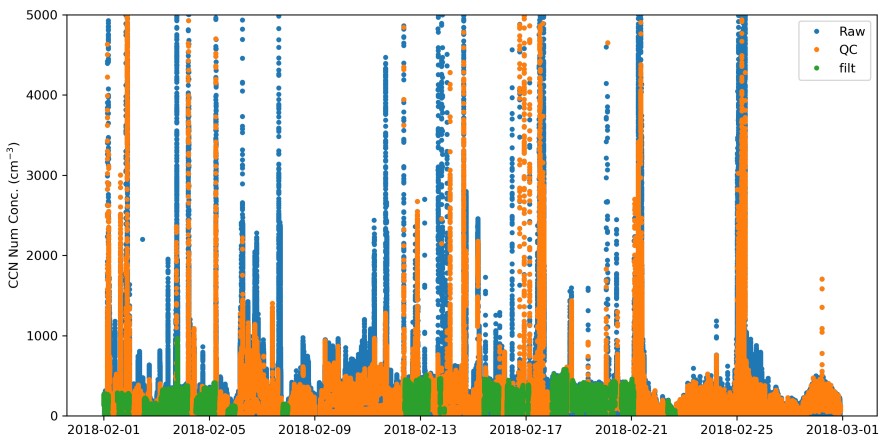

**Figure A2.** Exhaust contamination of CCN$_{0.5}$ data from February 2018 of the MARCUS voyage showing the amount of data remaining after quality control (orange), and the subsequent removal of exhaust contaminated data (green).

| | Latitude Bin ($^o$S) | Mean | Median | Count | Std. Dev | Min | Max | Percentiles $25^{th}$ | $75^{th}$ | $10^{th}$ | $90^{th}$ |
|---|---|---|---|---|---|---|---|---|---|---|---|
| $CCN_{0.2}$ | 40-45 | 169 | 125 | 110 | 148 | 12 | 812 | 49 | 270 | 36 | 345 |
| | 45-50 | 131 | 130 | 199 | 84 | 16 | 337 | 51 | 192 | 37 | 245 |
| | 50-55 | 102 | 87 | 235 | 86 | 13 | 1065 | 63 | 125 | 46 | 162 |
| | 55-60 | 102 | 99 | 185 | 52 | 12 | 238 | 53 | 145 | 39 | 174 |
| | 60-65 | 118 | 105 | 518 | 70 | 12 | 306 | 66 | 162 | 35 | 218 |
| | 65-70 | 133 | 123 | 456 | 61 | 27 | 299 | 97 | 177 | 51 | 226 |
| $CCN_{0.5}$ | 40-45 | 322 | 267 | 111 | 279 | 15 | 1416 | 105 | 443 | 41 | 674 |
| | 45-50 | 197 | 188 | 199 | 139 | 13 | 520 | 61 | 281 | 36 | 395 |
| | 50-55 | 154 | 122 | 249 | 137 | 25 | 1358 | 74 | 201 | 57 | 251 |
| | 55-60 | 155 | 157 | 189 | 76 | 16 | 374 | 98 | 201 | 56 | 249 |
| | 60-65 | 157 | 145 | 518 | 88 | 11 | 405 | 83 | 222 | 47 | 280 |
| | 65-70 | 232 | 240 | 485 | 90 | 66 | 606 | 186 | 271 | 108 | 322 |
| $CN_{10}$ | 40-45 | 681 | 623 | 124 | 391 | 180 | 2512 | 388 | 839 | 333 | 1055 |
| | 45-50 | 366 | 282 | 175 | 218 | 76 | 965 | 224 | 499 | 153 | 725 |
| | 50-55 | 405 | 359 | 251 | 221 | 45 | 1128 | 220 | 529 | 176 | 691 |
| | 55-60 | 330 | 322 | 195 | 112 | 43 | 636 | 258 | 402 | 204 | 477 |
| | 60-65 | 306 | 300 | 512 | 143 | 44 | 924 | 202 | 393 | 111 | 483 |
| | 65-70 | 447 | 367 | 472 | 272 | 170 | 2013 | 298 | 446 | 271 | 764 |

**Table A1.** Statistics for $CCN_{0.2}$, $CCN_{0.5}$ and $CN_{10}$ for each of the latitude bins using data from both MARCUS and CAPRICORN2.

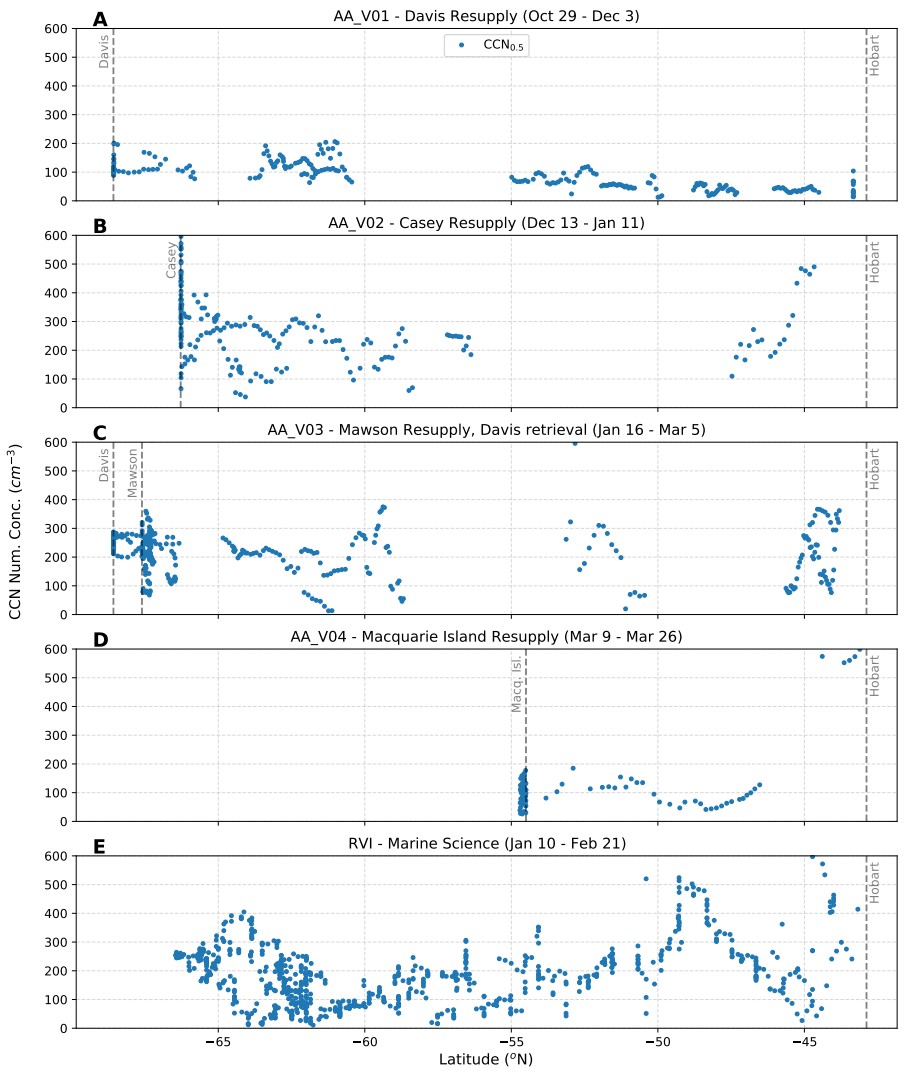

**Figure A3.** Latitudinal distributions of individual hourly data points for $CCN_{0.5}$, split into each of the voyages utilised in this study: A) MARCUS V1; B) MARCUS V2; C) MARCUS V3; D) MARCUS V4; E) CAPRICORN2 IN2018_V01. All data is exhaust filtered.

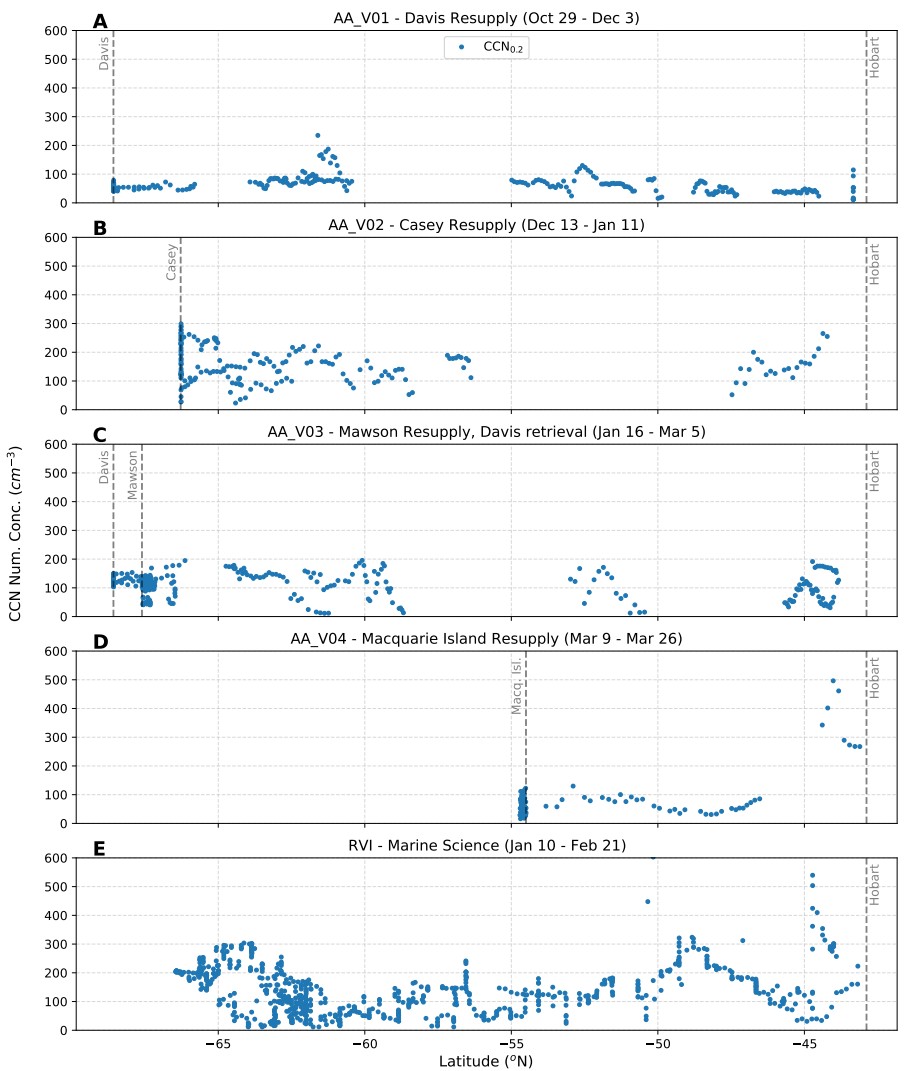

**Figure A4.** As in Figure A3, but for $CCN_{0.2}$ data

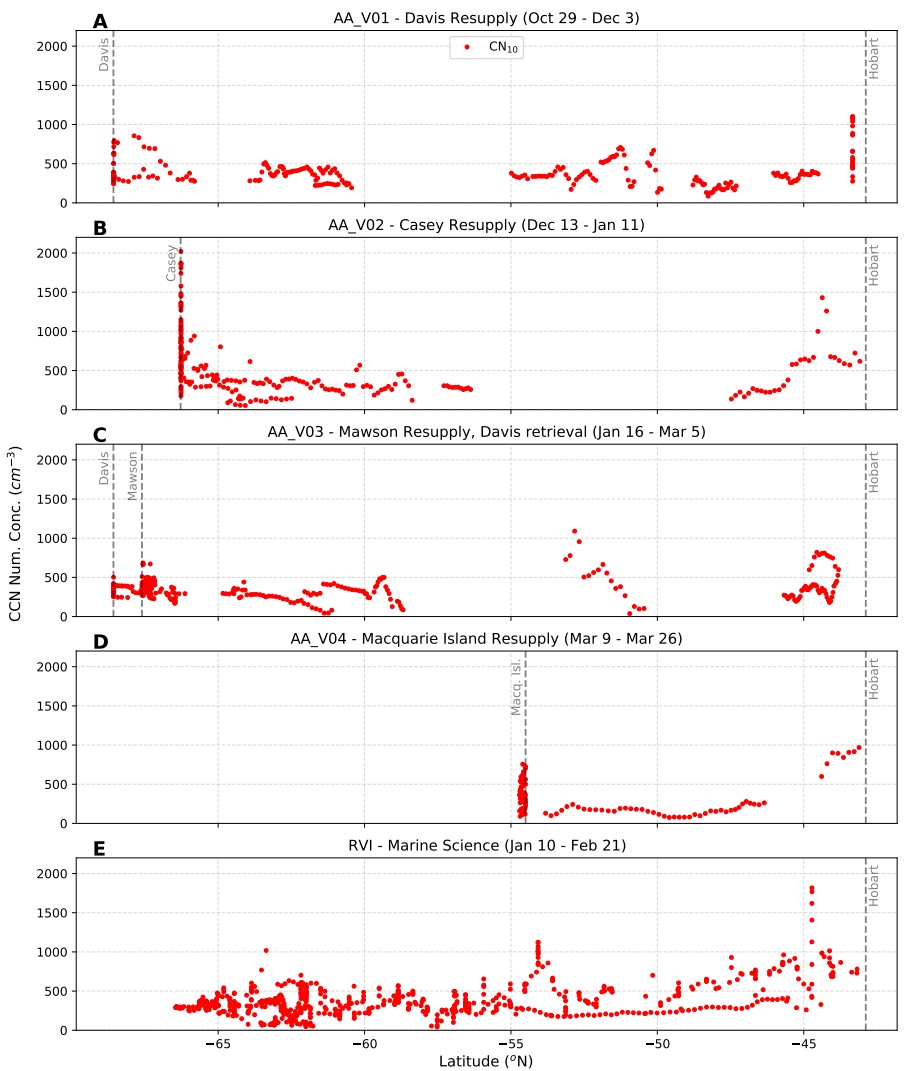

**Figure A5.** As in Figure A3, but for $CN_{10}$ data.

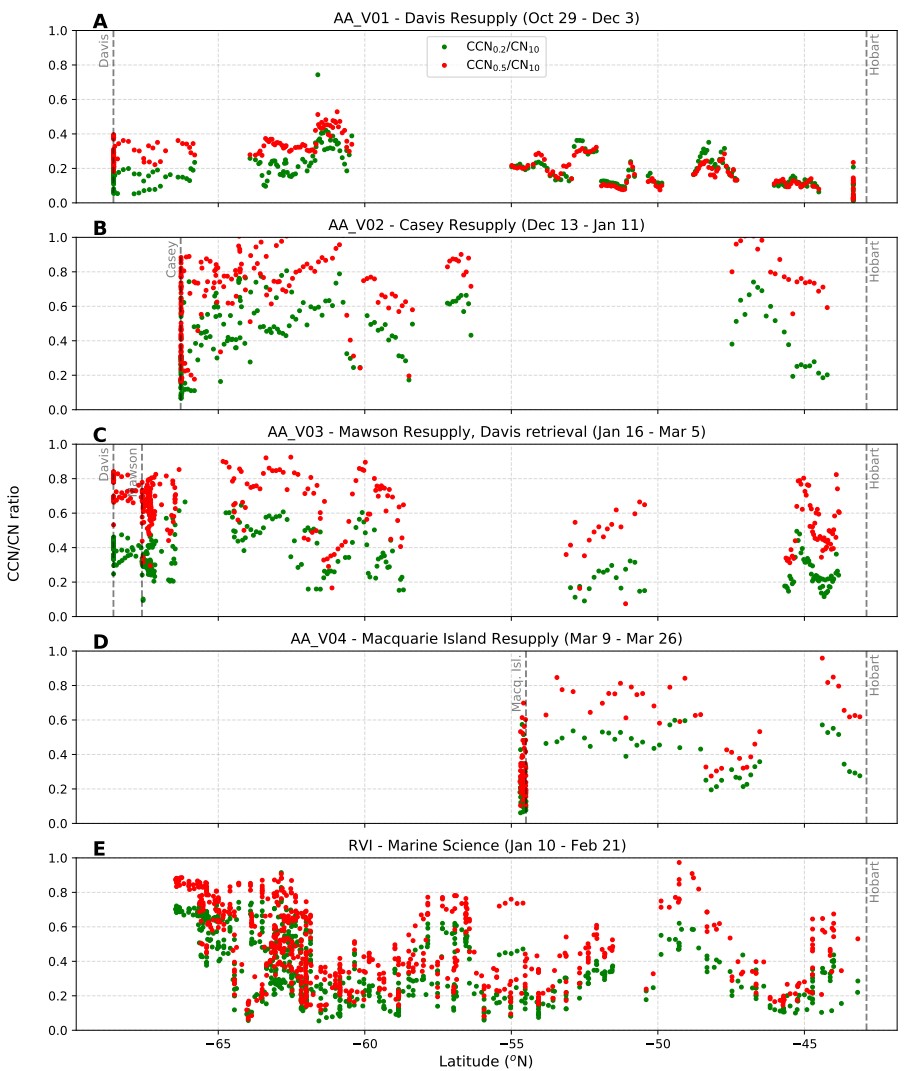

**Figure A6.** As in Figure A3, but for CCN/CN$_{10}$ ratio data

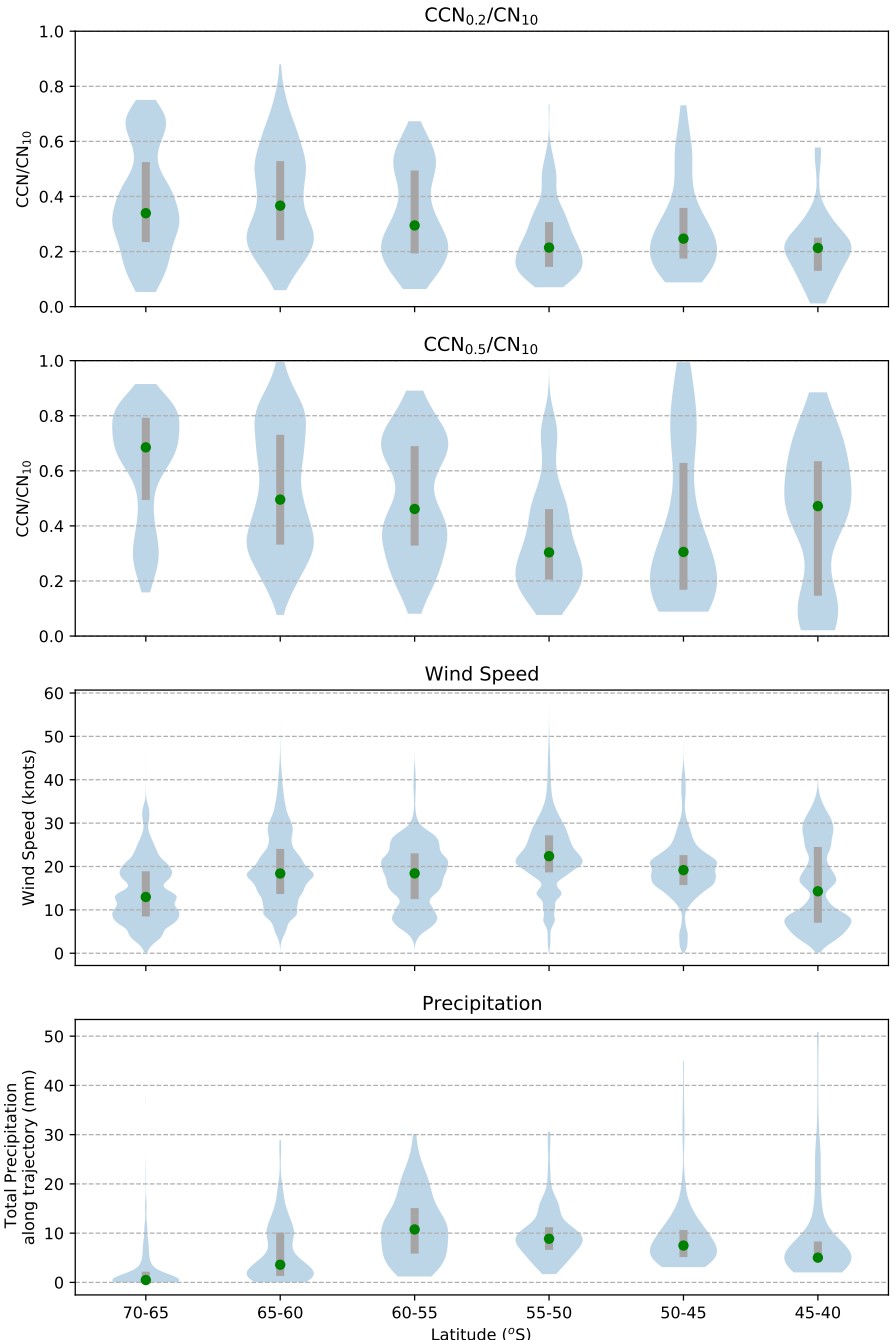

**Figure A7.** Latitudinal gradients of CCN/CN ratios (A and B), wind speed (C) and precipitation (D) using data from both MARCUS and CAPRICORN2.

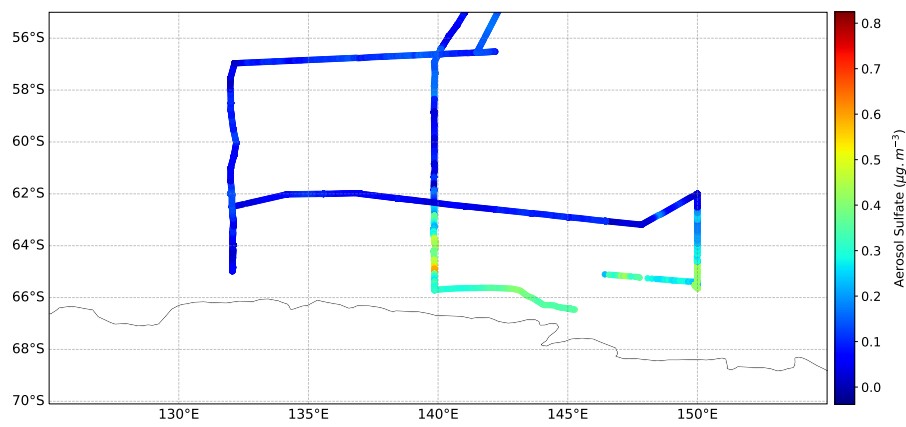

**Figure A8.** Voyage map from CAPRICORN2 showing the concentration of sulfate aerosol as measured by the ToF-ACSM as an indicator of the crossing of the Antarctic Atmospheric Compositional Front

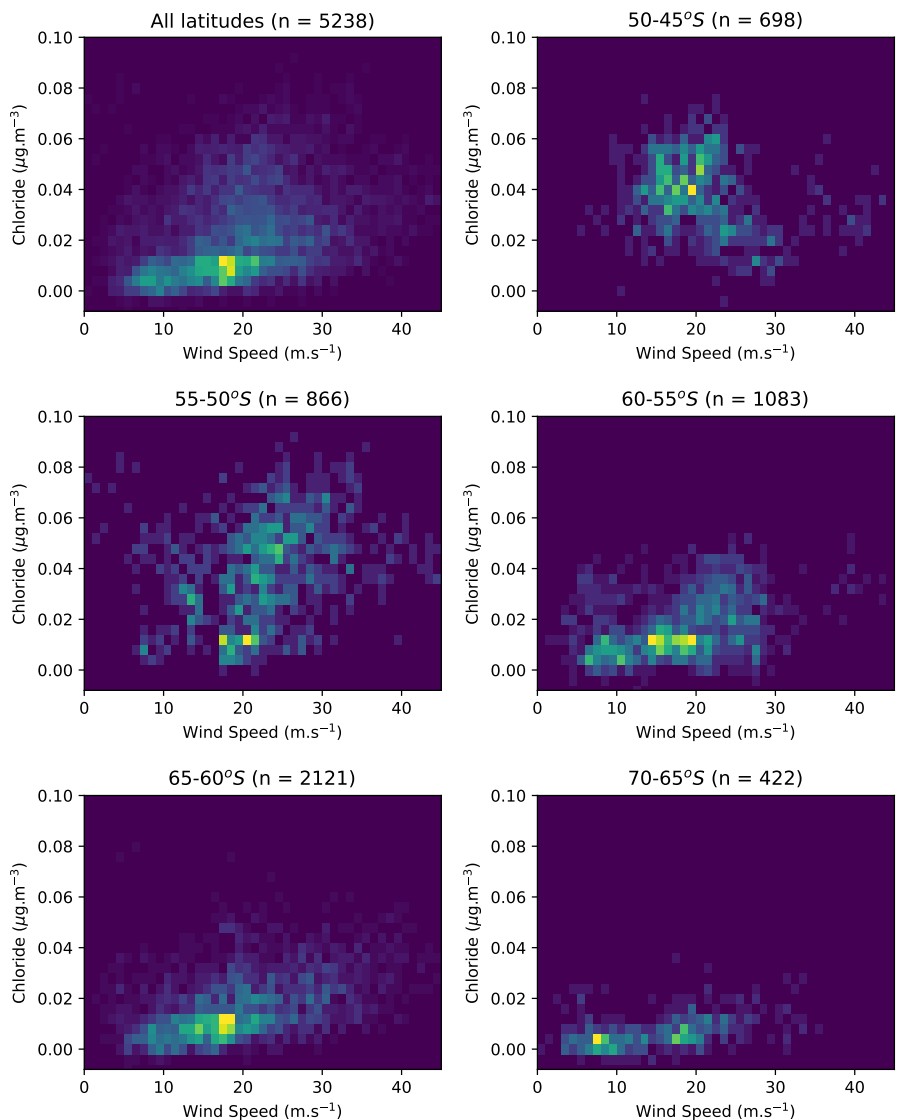

**Figure A9.** Two-dimensional histogram showing the positive correlation between wind speed and chloride (a tracer for seasalt measured with the tof-ACSM) across different latitude bands

*Author contributions.* RH undertook the CAPRICORN2 measurements, data quality control, analysis and interpretation. RH and MK wrote the manuscript. SG and RH undertook the trajectory analyses. RH, IM, JP, PS, ST, JH and MH were all critical in producing measurements from CAPRICORN2. CF and GK were PIs of the MARCUS CCN data. MK, JW and JH are responsibly for Cape Grim data, while RH, MK, JW and JH are responsible for Macquarie Island data. AP, SA and GM were chief scientists enabling all the measurements undertaken during CAPRICORN2, Macquarie Island (MICRE) campaigns, and MARCUS, respectively. GM contributed to the analysis and interpretation of data. All authors contributed to the writing and review of the manuscript.

*Competing interests.* The authors declare no competing interest.

*Acknowledgements.* MARCUS data were obtained from the Atmospheric Radiation Measurement (ARM) Program sponsored by the U.S. Department of Energy, Office of Science, Office of Biological and Environmental Research, and Climate and Environmental Sciences Division. We thank all the ARM technicians who collected the radiosonde and other data onboard R/V Aurora Australis. The Authors wish to thank the CSIRO Marine National Facility (MNF) for their support in the form of sea time on RV Investigator and associated support personnel, scientific equipment and data management during CAPRICORN2. In particular we thank Arron Tyndall, Francis Chui, Peter Shanks and Tegan Sime for their technical, IT and logistical support on board. Technical, logistical, and ship support for MARCUS were provided by the Australian Antarctic Division through Australia Antarctic Science projects 4292 and 4387 and we thank Steven Whiteside, Lloyd Symonds, Rick van den Enden, Peter de Vries, Chris Young and Chris Richards for assistance. The authors would also like to acknowledge the Australian Bureau of Meteorology for their long term and continued support of the Cape Grim Baseline Air Pollution Monitoring Station, and all the staff from the Bureau of Meteorology and CSIRO. We acknowledge the NASA Goddard Space Flight Center, Ocean Biology Processing Group, for the provision of the MODIS Aqua chlorophyll data and the NOAA Air Resources Laboratory (ARL) for the provision and support of the HYSPLIT transport and dispersion model.

This research was supported in part by BER Award DE-SC0018995 (GM and RH) and NASA grants 80NSSC19K1251 (GM). The work of GMM was funded by the United States Department of Energy Awards DE-SC0018626 and DE-SC0021159. GK acknowledges support from the Office of Science of the U.S. Department of Energy (DOE) as part of the Atmospheric System Research Program. The work of AP was partly funded by the National Environmental Science Program (NESP), Australia. This project received grant funding from the Australian Government as part of the Antarctic Science Collaboration Initiative program (RH, MK, AP, SA).

Funding for these voyages was provided by the Australian Government and the U.S. Department of Energy. All data and samples acquired on the voyages are made publicly available in accordance with MNF and AAD Policies. Raw data from MARCUS (Kuang et al., 2018; Kulkarni et al., 2018) are available from the DOE ARM's Data Discovery at https://adc.arm.gov/discovery/, Humphries et al. (2021a) and Humphries (2020).

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
