# Peer review of "Southern Ocean latitudinal gradients of Cloud Condensation Nuclei"

_Atmospheric Chemistry and Physics, 2020_

## Author Response (AR1)

Author responses to reviewer comments

ACP-2020-1246

"Southern Ocean latitudinal gradients of Cloud Condensation Nuclei"
by Ruhi S. Humphries et al., Atmos. Chem. Phys. Discuss.,
https://doi.org/10.5194/acp-2020-1246-RC1, 2021

**Referee #1**

We would like to thank reviewer 1 for the time taken to thoughtfully review our manuscript. The manuscript has been improved in important ways in response to their comments, details of which are described below in the format used in the reviewer's comments.

- Line 18 – we have edited this sentence to highlight the importance and need for more long-term stations.
- Line 38 and following – we thank the reviewer for highlighting this important paper. We have added it in as a reference.
- Line 195 and following – the vent emissions from indoor air and cooking, and more significantly from waste incineration, are important considerations. However, these emission sources are collocated with the exhaust emissions on both platforms, and represent a similar signal to the diesel exhaust itself, so are captured by the removal process described in the paper. We have clarified this in the first paragraph of the platform exhaust section.
- Line 315 – We have extended this paragraph to include the important results of Vallina 2006 and Fossum et al. 2020.
- Line 395 – we have altered the sentence to be more specific about the change expected in meteorological variables. We have also reviewed Bigg 1990 and Wylie et al. 1993 and while important pieces of work, we feel the main results are out of the scope of the current manuscript.

Technical/Minor revisions

- Line 212 – accepted and changed.
- Line 251 – Bandwidth selection is key to having the smoothed KDE be an accurate representation of the underlying data. Scott's factor is a widely accepted method for calculating the optimal bandwidth of a KDE. The inclusion of this sentence is just making it explicit how the bandwidth was calculated for these data. No changes have been made.
- Line 282 – accepted and changed.
- Line 299 and 389 – we have removed the word "step-change" in line 299, and "step" in line 389, as this concept is better discussed in other parts of the paragraphs and labelling it as a step could be misconstrued.
  We have added a sentence around line 295 to comment on the difference between $CCN_{0.5}$ and $CCN_{0.2}$ in that latitudinal bin.
  We have added a sentence around line 300 discussing our findings against Andreae 2009. We thank the reviewer for highlighting this important result.
- Line 300 – Thank you to the reviewer for picking up this issue. We have changed the figure referencing in that sentence to avoid confusion.
- Line 400 – accepted and changed.
- Line 412 – accepted and changed.

- Figure A7 – this was an issue with the plotting script where the axes labels were not plotted correctly. This has been rectified and Figure A7 has been updated.

**Referee #2**

We would like to thank reviewer 2 for their considered comments. They have been an asset to the study and have led to an improved manuscript. We have responded below to each of the comments in a format consistent with the reviewer's own submission.

Minor revisions

1. Figure 3 – the violin plots already included in Figure 3 are themselves a version of a histogram. Consequently, since the information an histogram would show is already included in the Figure, we have not added a separate histogram to the manuscript.
2. Figure 4 – we have limited the back trajectories to 5 days because of the uncertainty of trajectories in this region of the world. This was discussed in more detail in Humphries et al. 2016 (10.5194/acp-16-2185-2016).
3. Figure A3 and A4 – these figures, along with Figures A5 and A6 have been updated according to the reviewer's suggestions.
4. Figure A8, adding number of data points along the ship-track – this information is already displayed to some degree in Figures 1 and 2. All data presented are hourly data and adding the number of data points along a track needs some form of binning, which is already performed in Figure 2. To the latter part of the reviewer's comment, sea spray production from the ship itself is not a major concern on the RV Investigator (CAPRICORN2 campaign) because of the position of the inlet ~16.6 m above sea level and as far fore on the ship as possible. Any whitewash plume produced by the ship itself will generally fall well below the inlet. For the Aurora Australis (MARCUS campaign), while the measurement inlet is much higher (~25 m above sea level), the position of the inlet further aft on the ship's superstructure means that the influence of these plumes on measurements is possible. However given our analysis is focussed on number concentrations, and most of the spray will be on the larger end of the aerosols sizes, their number will be relatively small, and is likely to be insignificant. In addition, the MARCUS data set only makes up a small proportion of the dataset due to the exhaust contamination, so it highly unlikely to change the conclusions of the paper.
5. Line 252 – this section has been expanded to indicate the variable names and temporal and spatial resolution of the data utilised.
6. Line 404
7. The limiting factor in composing a two dimensional histogram plot (similar to Figure A9) of chl-a vs CCN is that unlike sea salt aerosols, which are produced at the source, secondary aerosols formed from phytoplankton emissions take a number of days to form. So except for very rare conditions, the wind conditions in this part of the world, together with the time required for nucleation and growth, mean that precursor emissions from phytoplankton occur several thousand km upwind from where they could be measured as CN or CCN. As such, composing such a plot would not be informative, and could in fact be misleading.

Minor errors

1. Neither of lines 1 or 29 require alteration, as the noun of each sentence is clearly stated as "region" or "atmosphere", respectively.
2. Changed to the reviewer's suggestion.

3.  Changed to the reviewer's suggestion.
4.  Field campaign abbreviations have been expanded based on reviewer's suggestion.
5.  Expanded based on reviewer's suggestions.
6.  The reviewer's suggestion doesn't improve the sentence, so we have left it as is.
7.  This changes the topic being referred to, making it inaccurate. We have left it as is.
8.  The CLAW acronym is just the initials of the initial authors and expanding these doesn't add to the interpretation of the acronym. In this case, the CLAW hypothesis is referred to throughout the literature with the acronym, without expansion. We have left this as is.
9.  Changed to the reviewer's suggestion.
10. This sentence has been split into two to further elaborate on the key findings of Alroe et al. related to this paper.
11. Changed to the reviewer's suggestion.
12. Changed to the reviewer's suggestion.
13. This is a well-known and well-defined term in the domain, and we will leave it to the reader to look-up the definition where required.